# Multidrug-resistant and extended-spectrum beta-lactamase-producing Enterobacteriaceae isolated from chicken droppings in poultry farms at Gondar City, Northwest Ethiopia

**Mitkie Tigabie** [ORCID]*, **Sirak Biset** [ORCID], **Teshome Belachew, Azanaw Amare, Feleke Moges**

Department of Medical Microbiology, School of Biomedical and Laboratory Sciences, College of Medicine and Health Sciences, University of Gondar, Gondar, Ethiopia

* mitku1621@gmail.com

**Data Availability Statement:** All relevant data are within the manuscript and its Supporting Information files

## Abstract

### Background

The poultry sector is one of the largest and fastest-growing agricultural sub-sector, especially in developing countries like Ethiopia. In poultry production, poultry farmers use sub-optimum doses of antibiotics for growth promotion and disease prevention purpose. This indiscriminate use of antibiotics in poultry farms contributes to the emergence of antibiotic-resistant bacteria, which has adverse implications for public health. Therefore, this study is aimed to assess multidrug resistance and extended-spectrum beta-lactamase-producing Enterobacteriaceae from chicken droppings in poultry farms.

### Methods

A total of 87 pooled chicken-dropping samples were collected from poultry farms from March to June 2022. Samples were transported with buffered peptone water. Selenite F broth was used for the enrichment and isolation of *Salmonella* spp. Isolates were cultured and identified by using MacConkey agar, Xylose lysine deoxycholate agar, and routine biochemical tests. Kirby-Bauer disk diffusion technique and combination disk test were used for antibiotic susceptibility testing and confirmation of extended-spectrum beta-lactamase production, respectively. Data were entered using Epi-data version 4.6 and then exported to SPSS version 26 for analysis.

### Result

Out of 87 pooled chicken droppings, 143 Enterobacteriaceae isolates were identified. Of these, *E. coli* accounts for 87 (60.8%), followed by *Salmonella* spp. 23 (16.1%), *P. mirabilis* 18 (12.6%) and *K. pneumoniae* 11 (7.7%). A high resistance rate was observed for ampicillin 131 (91.6%), followed by tetracycline 130 (90.9), and trimethoprim-sulfamethoxazole 94 (65.7%). The overall multidrug resistance rate was 116/143 (81.1%; 95% CI: 74.7–87.5). A

**Funding:** The authors have not received any specific funding for this work.

**Competing interests:** The authors have declared that no competing interests exist.

**Abbreviations:** ABR, Antibiotics Resistance; AST, Antibiotic Susceptibility Testing; ATCC, American Type Culture Collection; BPW, Buffered Peptone Water; CLSI, Clinical and Laboratory Standards Institute; ESBL, Extended Spectrum Beta-Lactamase; MDR, Multidrug Resistance.

total of 12/143 (8.4%; CI: 3.9–12.9) isolates were extended-spectrum beta-lactamase producers, with 11/87 (12.6%) *E. coli* and 1/11 (9.1%) *K. pneumoniae*.

## Conclusion and recommendations

High prevalence of multi-drug resistant isolates was observed. This study alarms poultry as a potential reservoir of extended-spectrum beta-lactamase-producing Enterobacteriaceae, which might shed and contaminate the environment through faecal matter. Prudent use of antibiotics should be implemented to manage antibiotic resistance in poultry production.

## Introduction

Poultry is one of the most widespread food animals and chicken is the largest farmed animal species worldwide [1]. The poultry sector is one of the largest and fastest-growing agricultural sub-sector, especially in developing countries like Ethiopia. It is an essential component of the country's economy, providing income for farmers and a good source of high-quality protein for the ever-growing population of Ethiopia [2]. However, in the poultry sector, in addition to using antibiotics for therapy and disease prevention, antibiotics are regularly added to poultry feed in sub-therapeutic doses for growth promotion [3].

Globally, over 50% of antibiotics are used by the food animal industry and an increase of 50% in antibiotic usage for farming is estimated by 2030 [4]. An estimated 25 million pounds of antimicrobials are used for non-therapeutic purposes in chickens, pigs, and cows, while only 3 million pounds are used for human medicine worldwide [5].

Developed countries have implemented prudent antibiotic use policies and surveillance systems both in clinical and veterinary settings. There are no such systems in low and middle-income countries [3]. In these countries, antibiotics are used in poultry for three main reasons: 1) Poultry flock treatment when illness is first recognized in a small proportion of the chickens; 2) to prevent diseases when the physical stress involved in the movement of chickens in large numbers; and 3) as a growth promoter to boost chickens weight [6, 7].

The irrational use of antibiotics in poultry farms for growth promotion and disease prevention triggers high selection pressure among bacterial agents, which might contribute to the emergence and development of antibiotic-resistant (ABR) bacteria [8]. Antibiotic resistance increases time-to-time. It has been declared by World Health Organization as one of the top ten global public health threats in the 21st century [9]. Currently, an estimated 700,000 people a year die of ABR infections in the globe. If action is not taken, this number could rise to around 10 million per year, with a global loss of 100 trillion United States dollars by 2050 [10]. More than 2.8 million ABR occur, resulting in more than 35,000 deaths annually in the United States alone [11]. In Africa, approximately 4.2 million deaths also occur annually due to ABR [10].

Extended-spectrum beta-lactamase (ESBL) genes have led to the emergence of bacteria that are resistant to most antibiotics [12]. Extended-spectrum beta-lactamase is an enzyme that can hydrolyze penicillin, cephalosporins, and aztreonam and is inhibited by beta-lactamase inhibitors, like clavulanic acid [13]. The most common ESBL types found in poultry and poultry products are CTX-M-1, TEM-52 and SHV-12. Extended-spectrum beta-lactamase-producing bacteria are also, present in every type of commercial chicken and can be detected even in newly hatched chickens. This enzyme is most common in gram-negative bacteria, particularly in Enterobacteriaceae such as *Escherichia coli* (*E. coli*) and *Klebsiella pneumoniae* (*K.*

*pneumoniae*) [12]. Some of these bacteria are significant causes of foodborne, urinary tract, respiratory tract, bloodstream, and wound infections in humans [14].

Extended-spectrum beta-lactamase-producing bacteria in the poultry sector are recognized as a potential community health concern. Because, it can be transmitted through food chains, in close contact with poultry, leafy vegetables and via bodies of water contaminated with poultry droppings. So, it is very important to monitor the resistance to antibiotics not only in human bacterial pathogens but also in pathogenic and commensal bacteria of poultry origin [15–17].

Awareness of the prevalence of ABR in poultry provides baseline data to implement an integrated ABR surveillance system and also facilitates the evaluation of interventions used to control the ABR. Monitoring and surveillance of ABR at poultry farms may help to reduce the transfer of ABR bacteria from poultry to humans directly or indirectly through the environment [18]. In Ethiopia, Multidrug resistance has not been well-studied and extended-spectrum beta-lactamase-producing Enterobacteriaceae from poultry droppings are still missing, particularly in Northwest Ethiopia. Therefore, this study is aimed to determine the multidrug resistance (MDR) and ESBL-producing Enterobacteriaceae from chicken droppings in poultry farms at Gondar City, Northwest Ethiopia.

## Materials and methods

### Study design, period, and area

A survey was conducted from March 1, 2022, to June 30, 2022. The study was conducted in Gondar City Ethiopia. Gondar is one of the ancient historical cities in Ethiopia and is located 737 Km from Addis Ababa, the country's capital. The city's total population is estimated to be 395,138 [19]. According to the information obtained from Gondar city's rural and urban agriculture centre, 87 poultry farms supply chickens and eggs to the society.

### Data collection and analysis

Data related to general characteristics and antibiotic use in the poultry farms were collected by face-to-face interview technique from the chicken caregivers or owners using a semi-structured questionnaire before sample collection. All data were collected and analyzed by a trained laboratory technologist.

### Sample collection, transportation, processing, and identification

A total of 87 chicken-dropping samples were randomly collected from poultry farms. A sample consisted of a pool of five fresh chicken droppings obtained from the five different parts of the poultry building [20].

Each farm was visited once, and the samples were collected using sterile applicator sticks and stored in sterile universal sampling bottles containing 90 ml buffer peptone water (BPW) (Himedia, India M614). A code was attributed to each universal sampling bottle and placed in a cooler (icebox) containing ice packs. Immediately, samples were transported to the School of Biomedical and Laboratory Sciences, Medical Microbiology laboratory section.

After homogenization, about 1 millilitre of the sample was further transferred into two different test tubes containing 9 ml of BPW (Himedia, India, M614) and 5 ml of selenite F broth (Himedia, India M414). Test tubes were incubated at 37˚C for 18–24 hrs. After incubation, samples from BPW were streaked on a MacConkey agar plate (Oxoid Ltd, Basingstoke, United Kingdom (UK)). Samples from selenite F broth were streaked on a xylose lysine deoxycholate

agar plate (XLD) (HiMedia, India, M608) [21]. All the plates were incubated aerobically at 37˚C for 24 hrs.

At the end of incubation, the MacConkey and XLD agar plates were examined for growth and preliminary identification of the bacteria was done based on the characteristics of the bacteria colony (size, shape, colour, texture, elevation, edge). In addition, the smear was prepared from each colony observed on the plates and gram staining was performed. The gram reaction and the shape of the bacteria were observed using a microscope.

After the identification of gram-negative bacteria, a series of biochemical tests were performed on colonies from pure cultures of the isolates. Triple sugar iron agar (TSI) (Oxoid Ltd, Basingstoke UK), Simon's citrate agar (Oxoid Ltd, Basingstoke, UK), urease agar (Oxoid Ltd, Basingstoke, UK), lysine iron agar (Oxoid Ltd, Basingstoke, UK) (LDC), and Sulphur indole motility medium (SIM) (Oxoid Ltd, Basingstoke, UK) were included in the biochemical tests for species identification [22].

## Antibiotic susceptibility testing

Following bacterial identification, the antibiotic susceptibility testing (AST) of the isolates was performed by a Kirby-Bauer disk diffusion technique. The colonies of a young culture were picked from the pure culture using a sterile wire loop and emulsified in 0.85% of normal saline to make bacterial suspension and compare with 0.5 McFarland turbidity standards. Then the bacterial suspension was inoculated onto Muller-Hinton agar (MHA) (Oxoid, Basingstoke, and Hampshire, UK) by lawn culture method. The AST was performed following the recommendation of the Clinical and Laboratory Standards Institute (CLSI) guideline 2021 against— ampicillin (10μ), gentamicin (10μg), tetracycline (30μg), nalidixic-acid [30] ciprofloxacin (5μg), chloramphenicol (30μg), trimethoprim-sulfamethoxazole (1.25μg/23.75μg), cefoxitin (30μg), cefotaxime (30μg), ceftazidime (30μg), ceftriaxone (30μg), and meropenem (10μg). All the antibiotic disks used were from BD, BBL[TM] Company, and USA Product. After overnight incubation at 37˚C for 16–18 hours, the zone of inhibition was measured by a ruler and the results was interpreted as resistant, intermediate, and sensitive [23]. Bacterial isolates that were resistant to at least one antibiotic agent in three or more antibiotic classes were considered MDR isolates [24].

## Detection of extended-spectrum beta-lactamase

All Enterobacteriaceae strains were tested against ceftriaxone, cefotaxime, and ceftazidime for ESBL screening using the Kirby-Bauer disk diffusion method. If the zone of inhibition was ≤ 22 mm for ceftazidime, ≤ 25 mm for ceftriaxone, and ≤ 27 for cefotaxime, they were considered as potential ESBLs-producing strains and selected for a further phenotypic confirmatory test as described below [23].

A phenotypic confirmatory test was done using a combined-disk diffusion test and interpreted by following the CLSI, 2021 guidelines. Pure culture of suspected ESBL producer isolates was emulsified in 0.85% saline and compared with 0.5 McFarland turbidity standard then inoculated on MHA by lawn culture method using sterile swabs. The following antibiotic disks such as cefotaxime (30μg), cefotaxime/clavulanic acid (30μg/10μg), ceftazidime (30μg), and ceftazidime/clavulanic acid (30μg/10μg) were used to confirm the status of the ESBL phenotypes. The plates were then incubated aerobically at 37˚C for 16–18 hrs. If greater or equal to 5mm an increase in zone diameter for cefotaxime and ceftazidime in combination with clavulanic acid than the zone diameter of the tested alone, it was confirmed as ESBL-producing isolates [23].

### Quality control

All culture media was prepared according to the manufacturer's instructions and following standard operational procedures. The sterility of newly prepared culture media was checked by incubating 5% of prepared culture media at 35–37˚C overnight before use and was evaluated for possible growth or contamination. The performance testing was performed with inoculating known control strains of *E. coli* American Type Culture Collection (ATCC) 25922 and *Salmonella Typhimurium* ATCC 14028 on culture media. For the ESBL confirmatory test, *K. pneumoniae* ATCC 700603 (ESBLs positive) and *E. coli* ATCC 25922 (ESBLs negative control) strains were used to check the quality of the culture media and antibiotic disks [23].

### Data processing and analysis

All data were checked for completeness, coded, and entered using Epi-data version 4.6 and the data was exported to Statistical Package for Social Sciences version 26 for further analysis. Frequency analysis was carried out to determine the frequency of independent variables and the prevalence of MDR isolates. Fisher's exact test was used to observe an appropriate association between independent variables and ESBL-producing isolates. A p-value of less than 0.05 at a 95% confidence interval in fisher's exact test was considered an association between independent variables and ESBL-producing isolates. The results were presented in texts, figures, and tables.

### Ethical approval

Ethical clearance was obtained from the Ethical Review Committee of the School of Biomedical and Laboratory Sciences, College of Medicine and Health Sciences, the University of Gondar with protocol reference number SBMLS/202, dated 14 February 2022. The owner of each poultry farm was informed about the aim of the study and oral permission was obtained from the owners/ managers before sampling.

## Results

### General characteristics of the poultry farms

A total of 87 poultry farms were visited, and a farm owner or chicken caregiver was interviewed about the farm's characteristics and how to handle the chickens. The majority of the poultry farms raised eggs layer chickens 55 (63.2%), used deep litter chicken housing systems 82 (94.3%), and used commercially prepared feeds 80 (92%). In most farms, 78 (89.7%) were not clean from chicken droppings and remained so until a new flock was introduced. In more than half of the poultry farms, diseased chickens weren't isolated and separated. Almost all of the farm owners or the people who looked after the chickens had no profession related to the poultry industry (Table 1).

### Antibiotic use in the poultry farms

Of most poultry farmers 82 (94.3%) used antibiotics on their farms. Antibiotics given in poultry farms were enrofloxacin, oxytetracycline, ciprofloxacin, trimethoprim and sulphadiazine. The majority of poultry farms 75/82 (91.5%) used antibiotics for both preventive and treatment purposes. Out of antibiotic users, most of the poultry farmers purchased their antibiotics from a veterinary pharmacy and gave them to their chickens by mixing them with feed or water (Table 2).

**Table 1. General characteristics of the poultry farms at Gondar City, Northwest Ethiopia, March to June 2022.**

| Variables | Category | Frequency N (%) | ESBL-status | | Fisher's Exact test p-value |
|---|---|---|---|---|---|
| | | | ESBL-positive | ESBL-Negative | |
| Type of commercial chicken | Layer | 55 (63.2) | 10 (18.2) | 45 (81.8) | 0.129 |
| | Broiler | 6 (6.9) | 0 (0) | 6 (100) | |
| | One day old | 26 (29.9) | 1 (3.8) | 25 (96.2) | |
| Flock size (number of chickens on the farm) | <500 | 48 (55.2) | 4 (8.3) | 44 (91.7) | 0.014* |
| | 500–1000 | 27 (31.0) | 2 (7.4) | 25 (92.6) | |
| | >1000 | 12 (13.8) | 5 (41.7) | 7(58.3) | |
| Age of chicken (months) | <2 | 34 (39.1) | 6 (17.6) | 28 (82.4) | 0.037* |
| | 2–6 | 14 (16.1) | 1 (7.1) | 13 (92.9) | |
| | 7–12 | 31 (35.6) | 1(3.2) | 30 (96.8) | |
| | >12 | 8 (9.2) | 3 (37.5) | 5 (62.5) | |
| Farm age (years) | <5 | 80 (92.0) | 8 (10) | 72 (90) | 0.040* |
| | 5–10 | 7 (8.0) | 3 (42.9) | 4 (57.1) | |
| Chicken housing system | Deep litter system | 82 (94.3) | 10 (12.2) | 72 (87.8) | 0.50 |
| | Traditional housing | 5 (5.7) | 1 (20) | 4 (80) | |
| Cleaning of chicken droppings | When the flock changed (the flock out) | 78 (89.7) | 11 (14.1) | 67 (85.9) | 0.278 |
| | By six months per a year | 9 (10.3) | 0 (0.0) | 9 (100) | |
| Timely isolation and separation of diseased chickens | Yes | 35 (40.2) | 8 (22.9) | 27 (77.1) | 0.024* |
| | No | 52 (59.8) | 3 (5.8) | 49 (94.2) | |
| Professional short-term training is given | Yes | 81 (93.1) | 10 (12.3) | 71 (87.7) | 0.567 |
| | No | 6 (6.9) | 1 (16.7) | 5 (83.3) | |
| Owners and chicken caregiver profession is related to the chicken farm | Yes | 4 (4.6) | 1 (25.0) | 3 (75.0) | 0.424 |
| | No | 83 (95.4) | 10 (12.0) | 73 (88.0) | |
| Waste disposal | Send to field | 82 (94.3) | 10 (12.2) | 72 (87.) | 0.500 |
| | Compost | 5 (5.7) | 1 (20.0) | 4 (80.0) | |
| Feeding condition | Commercially prepared | 80 (92.0) | 11 (13.8) | 69 (86.3) | 0.588 |
| | Both commercially and locally prepared | 7 (8.0) | 0 (0.0) | 7 (100) | |
| Water source | Well water | 20 (23.0) | 3 (15.0) | 17 (85.0) | 0.710 |
| | Pipe water | 67 (77.0) | 8 (11.9) | 59 (88.1) | |
| Chicken feeds contact with their droppings | Yes | 40 (46.0) | 2 (5.0) | 38 (95.0) | 0.058 |
| | No | 47 (54.0) | 9 (19.1) | 38 (80.9) | |

* Associations between independent variables and ESBL-producing isolates

## Prevalence of Enterobacteriaceae isolates from chicken droppings

Among a total of 87 poultry farms chicken-dropping samples 143 bacterial isolates were recovered. Of these, the most common isolates were *E. coli* 87 (60.8%), followed by *Salmonella* spp. 23 (16.1%), *P. mirabilis* 18 (12.6%) and *K. pneumoniae* 11 (7.7%). *E. coli* 87 (100%) was recovered from all samples collected. However, *Salmonella* spp. were isolated in 23 (26.4%; 95% CI:17.2–35.6) and *P. mirabilis* in 18 (20.7%; 95% CI:12.6–28.7) samples (Fig 1).

## Antibiotic resistance patterns of Enterobacteriaceae

Out of 143 Enterobacteriaceae isolates, the highest resistance rate was observed for ampicillin 131 (91.6%) followed by tetracycline 130 (90.9), trimethoprim-sulfamethoxazole 94 (65.7%),

**Table 2. Type of antibiotics use in the poultry farms at Gondar City, Northwest Ethiopia, March to June 2022.**

| Variables | Category | Frequency N (%) | ESBL-status | | Fisher's Exact test p-value |
| --- | --- | --- | --- | --- | --- |
| | | | ESBL-positive | ESBL-Negative | |
| Antibiotics use | Yes | 82 (94.3) | 11 (13.4) | 71(86.6) | 1.00 |
| | No | 5 (5.7) | 0 (0.0) | 5 (100) | |
| Use of enrofloxacin | Yes | 68 (82.9) | 11 (16.2) | 57(83.8) | 0.197 |
| | No | 14 (17.1) | 0 (0.0) | 14 (100) | |
| Use of oxytetracycline | Yes | 62 (75.6) | 10 (16.1) | 52 (83.9) | 0.279 |
| | No | 20 (24.4) | 1 (5.0) | 19 (95.0) | |
| Use of trimethoprim and sulphadiazine | Yes | 14 (17.1) | 5 (35.7) | 9 (64.3) | 0.018* |
| | No | 68 (82.9) | 6 (8.8) | 62 (91.2) | |
| Use of ciprofloxacin | Yes | 8 (9.8) | 5 (62.5) | 3 (37.5) | 0.001* |
| | No | 74 (90.2) | 6 (8.1) | 68 (91.9) | |
| Antibiotics used for treatment purposes | Yes | 71 (86.6) | 11 (15.5) | 60 (84.5) | 0.345 |
| | No | 11 (13.4) | 0 (0.0) | 11 (100) | |
| Antibiotics used for prevention purposes | Yes | 4 (4.9) | 0 (0.0) | 4 (100) | 1.00 |
| | No | 78 (95.1) | 11 (13.8) | 67 (85.9) | |
| Antibiotics are used for both prevention and treatment purposes | Yes | 75 (91.5) | 11 (14.1) | 64 (85.3) | 0.586 |
| | No | 7 (8.5) | 0 (0.0) | 7 (100) | |
| Frequency of antibiotics use | Regularly | 8 (9.8) | 1 (12.5) | 7 (87.5) | 1.00 |
| | Occasionally | 74 (90.2) | 10 (13.5) | 64 (86.5) | |
| Sources of antibiotics | Veterinary drug store | 73 (89) | 7 (9.6) | 66 (90.4) | 0.016* |
| | Parallel market | 9 (11) | 4 (44.4) | 5 (55.6) | |
| A common route of antibiotics administration | Mixed with feed and/or water | 76 (92.7) | 11 (14.5) | 65(85.5) | 1.00 |
| | Injection or others | 6 (7.3) | 0 (0) | 6 (100) | |

* Associations between independent variables and ESBL-producing isolates

and nalidixic acid 94 (65.7) and lowest resistance was observed against meropenem 13 (9.1%), gentamicin 16 (11.2%) and cefoxitin 24 (16.8%) (Table 3).

Regarding the resistance rate of individual bacterial isolates, *E. coli* demonstrated a high rate of resistance against ampicillin 80/87 (92.0%), tetracycline 79/87 (90.8%), nalidixic acid 59/87 (67%), and trimethoprim-sulfamethoxazole 55/87 (63.2%). Likewise, *K. pneumoniae* isolates showed a high resistance rate against ampicillin 11/11 (100%), tetracycline 10/11 (90.9%), trimethoprim-sulfamethoxazole 9/11 (81.8%), and nalidixic acid 7/11 (63.6%). All isolates showed a lower resistance rate against meropenem and gentamicin, with a range of 9.2% to 18.2% and 4.3% to 25.0%, respectively.

## Multi-drug resistant patterns of Enterobacteriaceae

A total of 12 antibiotics from 8 classes (aminoglycosides, amphenicol, carbapenems, cephalosporins, fluoroquinolones, folate pathway inhibitors, penicillin, and tetracycline) were used to assess the MDR patterns of isolates. The overall MDR prevalence in this study was 116/143 (81.1%; 95% CI: 74.7–87.5). The most common MDR isolates identified in this study were *E. coli* 73/87 (83.9%; 95% CI: 76.3–91.5) followed by *K. pneumoniae* 9/11 (81.8%; 95% CI: 69.1–94.5), *P. mirabilis* 14/18 (77.8%; 95% CI: 58.8–96.8), and *Salmonella* spp. 17/23 (73.9%; 95% CI: 55.9–91.9) (Table 4).

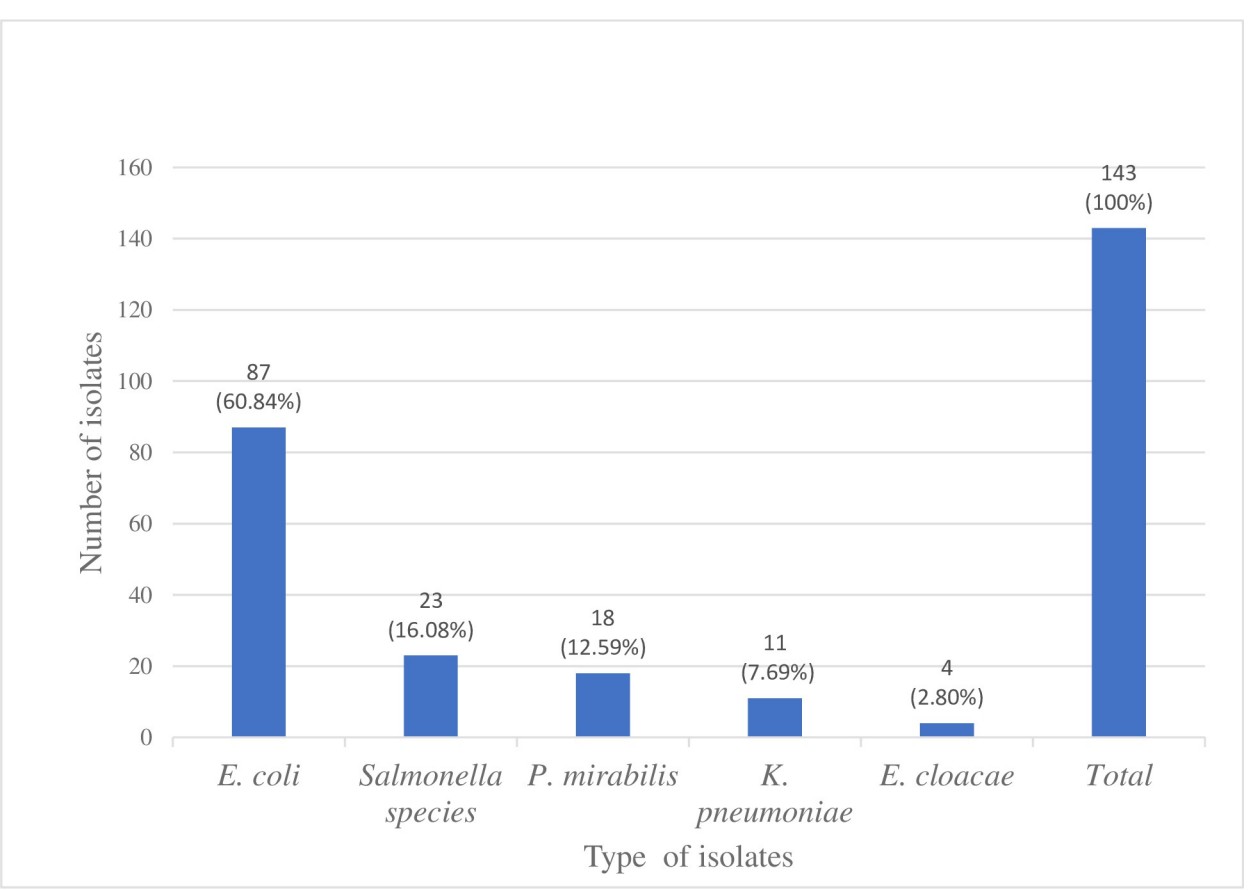

**Fig 1. The proportion of Enterobacteriaceae isolates from chicken droppings in poultry farms at Gondar City, Northwest Ethiopia, March to June 2022.**

### The prevalence of ESBL-producing Enterobacteriaceae

Among 143 bacterial isolates tested for ESBL, 12 (8.4%; CI: 3.9-12-9) were found to be positive. Of these, 11/87 (12.6%; 95% CI: 5.5–20.1) were *E. coli* and only one of the isolates was 1/11 (9.1%; 95% CI: 1.5–27.3) *K. pneumoniae.*

## Discussion

Antibiotic-resistant bacteria are a significant public health concern because the resistant bacteria and their mobile genetic elements disseminate among animals, humans, and the environment [25].

In this study, a total of 87 pooled chicken droppings were collected for bacteriological analysis and all of them were culture-positive. The culture-positivity rate in this study is in agreement with reports from Tanzania (100%) [26] and Indonesia (100%) [27]. However, it is higher than a study conducted in Jimma, Ethiopia 43.6% [28], Tanzania 55.2% [29], Egypt 12.5% and 25.6% [30, 31] Cameroon 44.1% [32], Nigeria 29.5% [33] and Albania 52.9% [34]. The difference in culture positivity rate may be due to the methods used to isolate the bacteria, the types of samples, and hygienic conditions in different places. In this study, for example, different types of samples were pooled, and most farms did not clean chicken droppings until a new flock was introduced, resulting in them being mixed with chicken feed, which fosters the cross-contamination of chickens [35].

**Table 3. Antibiotic resistance patterns of Enterobacteriaceae from chicken droppings in poultry farms at Gondar city, Northwest Ethiopia, March to June 2022.**

| Class | Antibiotics | *E. coli* | | *Salmonella* species | | *P. mirabilis* | | *K. pneumoniae* | | *E. cloacae* | | Total | |
|---|---|---|---|---|---|---|---|---|---|---|---|---|---|
| | | N = 87 | | N = 23 | | N = 18 | | N = 11 | | N = 4 | | N = 143 | |
| | | S | R | S | R | S | R | S | R | S | R | S | R |
| | | N (%) | N (%) | N (%) | N (%) | N (%) | N (%) | N (%) | N (%) | N (%) | N (%) | | N (%) |
| Aminoglycosides | GEN | 78 (89.7) | 9 (10.3) | 22 (95.7) | 1 (4.3) | 15 (83.3) | 3 (16.7) | 9 (81.8) | 2 (18.2) | 3 (75.0) | 1 (25.0) | 127 (88.8) | 16 (11.2) |
| Carbapenems | MER | 79 (90.8) | 8 (9.2) | 23 (100) | - | 15 (83.3) | 3 (16.7) | 9 (81.8) | 2 (18.2) | 4 (100) | - | 130 (90.9) | 13 (9.1) |
| Cephalosporins | CXT | 76 (87.4) | 11 (12.6) | 21 (91.3) | 2 (8.7) | 13 (72.2) | 5 (27.8) | 8 (72.7) | 3 (27.3) | 1 (25.0) | 3 (75.0) | 119 (83.2) | 24 (16.8) |
| | CAZ | 72 (82.8) | 15 (17.2) | 18 (78.3) | 5 (21.7) | 14 (77.8) | 4 (22.2) | 7 (63.6) | 4 (36.4) | 1 (25.0) | 3 (75.0) | 112 (78.3) | 31 (21.7) |
| | CRO | 75 (86.2) | 12 (13.8) | 19 (82.6) | 4 (17.4) | 15 (83.3) | 3 (16.7) | 8 (72.7) | 3 (27.3) | 3 (75.0) | 1 (25.0) | 120 (83.9) | 23 (16.1) |
| | CTX | 72 (82.8) | 15 (17.2) | 18 (78.3) | 5 (21.7) | 14 (77.8) | 4 (22.2) | 7 (63.6) | 4 (36.4) | 1 (25.0) | 3 (75.0) | 112 (78.3) | 31 (21.7) |
| Quinolones | NAL | 28 (32.2) | 59 (67.8) | 11 (47.8) | 12 (52.2) | 5 (27.8) | 13 (72.2) | 4 (36.4) | 7 (63.6) | 1 (25.0) | 3 (75.0) | 49 (34.3) | 94 (65.7) |
| | CIP | 68 (78.2) | 19 (21.8) | 20 (87.0) | 3 (13.0) | 7 (38.9) | 11 (61.1) | 8 (72.7) | 3 (27.3) | 2 (50.0) | 2 (50.0) | 105 (73.4) | 38 (26.6) |
| Penicillin | AMP | 7 (8.0) | 80 (92.0) | 5 (21.7) | 18 (78.3) | - | 18 (100) | - | 11 (100) | - | 4 (100) | 12 (8.4) | 131 (91.6) |
| Phenicol | CHL | 62 (72.3) | 25 (28.7) | 21 (91.3) | 2 (8.7) | 10 (55.6) | 8 (44.5) | 6 (54.5) | 5 (45.5) | 3 (75.0) | 1 (25.0) | 102 (71.3) | 41 (28.7) |
| Sulfonamides (folate pathway inhibitors) | SXT | 32 (36.8) | 55 (63.2) | 9 (39.1) | 14 (60.9) | 5 (27.8) | 13 (72.2) | 2 (18.2) | 9 (81.8) | 1 (25.0 | 3 (75.0) | 49 (34.3) | 94 (65.7) |
| Tetracycline | TET | 8 (9.2) | 79 (90.8) | 4 (17.4) | 19 (82.6) | - | 18 (100) | 1 (9.1) | 10 (90.9) | - | 4 (100) | 13 (9.1) | 130 (90.9) |

Key: S = Sensitive, R = Resistance, AMP = ampicillin; TET = tetracycline; SXT = trimethoprim-sulfamethoxazole; NAL = nalidixic acid; CHL = chloramphenicol; CIP = ciprofloxacin; CAZ = ceftazidime; CTX = cefotaxime, CRO = ceftriaxone; CXT = cefoxitin; GEN = gentamicin; MER = meropenem

In the current study, 143 Enterobacteriaceae isolates were identified, and the most predominant isolate was *E. coli* 87 (60.8%), followed by *Salmonella* spp. 23 (16.1%) and *P. mirabilis* 18 (12.6%). The same finding was also reported from Jimma, Ethiopia [28], Kenya [21], Nigeria [36], Côte d'Ivoire [20], and Malaysia [37]. The predominance of *E. coli* in this and many other studies may be because *E. coli* is a ubiquitous commensal bacterium that is predominantly found in the gastrointestinal tracts of animals and humans as a normal flora [38].

This study revealed that isolates from chicken droppings showed high resistance against ampicillin, tetracycline, and trimethoprim-sulfamethoxazole. This was also reported from Jimma, Ethiopia [28], Hawassa, Ethiopia [39], Tanzania [26, 29], Zambia [40], Cameroon [32], Côte d'Ivoire [20], Bangladesh [41], and Indonesia [42]. This demonstrates that these antibiotics are relatively cheap, easily accessible, and widely used antibiotics in the countries [43]. During farming, antibiotics are used for treatment or preventive purposes that favours the spread of ABR Enterobacteriaceae which can infect humans through the food chain [44]. In this study, these antibiotics were used in poultry for treatment or preventive purposes.

Moreover, the current study recorded higher resistance to quinolones like nalidixic acid 94 (65.7%) and ciprofloxacin 38 (26.6%). The use of quinolones for therapeutic purposes on the farm may be a possible contribution. The resistant pattern of Enterobacteriaceae in poultry to clinically important antibiotics in humans that are used for treating infections is a great

**Table 4. Multidrug resistance profiles of Enterobacteriaceae isolates from chicken droppings in poultry farms at Gondar city, Northwest Ethiopia, March to June 2022.**

| Resistance pattern | No. of antibiotics (classes) | Type of isolate N | | | | | |
|---|---|---|---|---|---|---|---|
| | | *E. coli* N = 87 | *K. pneumoniae* N = 11 | *P. mirabilis* N = 18 | *E. cloacae* N = 4 | *Salmonella* spp. N = 23 | Total N = 143 |
| Susceptible for all drug | – | 4 | - | - | - | 3 | 7 |
| TET | 1 (1) | 3 | - | - | - | 2 | 5 |
| AMP | 1 (1) | 4 | 1 | - | - | - | 5 |
| AMP TET | 2 (2) | 3 | 1 | 4 | 1 | - | 9 |
| AMP SXT | 2 (2) | - | - | - | - | 1 | 1 |
| AMP, TET, SXT | 3 (3) | 12 | 2 | 1 | - | 5 | 20 |
| AMP, TET, NAL | 3 (3) | 11 | - | 1 | - | 3 | 15 |
| AMP, TET, SXT, NAL | 4 (4) | 12 | - | - | - | 1 | 13 |
| AMP, TET, SXT, CHL | 4 (4) | 2 | - | - | - | - | 2 |
| AMP, TET, NAL, CIP | 4 (3) | 5 | - | - | - | 1 | 6 |
| AMP, TET, SXT, NAL, CIP | 5 (4) | - | - | 3 | - | 1 | 4 |
| AMP, TET, SXT, NAL, CHL | 5 (5) | 8 | 1 | 1 | - | 1 | 11 |
| AMP, TET, NAL, CIP, CXT | 5 (4) | 2 | - | - | - | - | 2 |
| AMP, TET, SXT, NAL, CIP, CHL | 6 (5) | 2 | - | - | - | - | 2 |
| AMP, TET, SXT, NAL, CHL, CXT | 6 (6) | 3 | 2 | - | - | - | 5 |
| AMP, TET, SXT, NAL, CIP, CHL, CXT | 7 (6) | 1 | - | 4 | - | - | 5 |
| AMP, TET, SXT, NAL, CAZ CTX, CRO | 7 (5) | 4 | 1 | - | - | 3 | 8 |
| AMP, TET, SXT, NAL, CIP CXT, CAZ, CTX | 8 (5) | 2 | 1 | 1 | 2 | 1 | 7 |
| AMP, TET, SXT, NAL, CIP, CHL, CAZ CTX, CRO, GEN, MER | 9 (6) | 6 | 2 | 3 | - | - | 11 |
| AMP, TET, SXT, NAL, CHL, CXT, CAZ CTX, CRO, GEN | 9 (6) | 1 | - | - | 1 | 1 | 3 |
| AMP, TET, SXT, NAL, CIP, CHL, CXT, CAZ CTX, CRO, GEN, MER | 10 (6) | 2 | - | - | - | - | 2 |
| **Total non-MDR isolates N (%)** | – | **14 (16.1%)** | **2 (18.2%)** | **4 (22.2%)** | **1 (25.0%)** | **6 (26.1%)** | **27 (18.9%)** |
| **Total MDR isolates N (%)** | – | **73 (83.9%)** | **9 (81.8%)** | **14 (77.8%)** | **3 (75.0%)** | **17 (73.9%)** | **116 (81.1%)** |

Key: AMP = ampicillin; TET = tetracycline; SXT = trimethoprim-sulfamethoxazole; NAL = nalidixic acid; CHL = chloramphenicol; CIP = ciprofloxacin; CAZ = ceftazidime; CTX = cefotaxime, CRO = ceftriaxone; CXT = cefoxitin; GEN = gentamicin; MER = meropenem; MDR = multidrug-resistant (against ≥3 antimicrobial classes)

concern [45]. For instance, in this study, enrofloxacin and ciprofloxacin are the most used antibiotics in poultry farms.

Bacterial isolates in the present study showed a relatively lower rate of resistance against meropenem 13 (9.1%) and gentamicin 16 (11.2%). This finding is supported by other studies, in Hawassa, Ethiopia [39], Kenya [21], and Albania [34]. Those studies reported 0% to 15% resistance for gentamicin and meropenem. Also, in Tanzania, gentamicin 10.3%, and 10.8% [26, 29], However, our result is lower than the studies conducted in Zambia, gentamicin 37.7% [40], Cameroon, meropenem 45% [32], Côte d'Ivoire, gentamicin 47.2% [20], Bangladesh, gentamicin 53% [41], and Indonesia, gentamicin 37% [27]. The possible explanation for a lower rate of resistance could be because of the inaccessibility of antibiotic agents that may not be given to poultry in the study area.

The prevalence of MDR *E. coli* in this study was 73 (83.9%). This finding is consistent with reports from Tanzania 86.8% [26], Zambia 85.7% [40], and Cameroon 83.1% [32] and lower

than the study in Albania 95% [34], and in Malaysia 100% [37]. However, it is higher than a study conducted in Jimma, Ethiopia 54.2% [28], Tanzania 69.3% [29], and Egypt 57.8% [30]. This bacteria strain might be human pathogenic *E. coli* since, similar virulence factors with the same mechanism between avian pathogenic *E. coli* and human extra-intestinal pathogenic *E. coli* strains [46], and genetic similarity between *E. coli* involved in urinary tract infections in humans and those found in poultry and poultry products has been demonstrated [47].

The multidrug resistance rate of Salmonella spp. was 17 (73.9%). This is in agreement with a report from Debre Zeit, Ethiopia 86.0% [48], Albania 82% [34], and Malaysia 82% [37]. However, it is higher than a study conducted in Jimma, Ethiopia 44.4% [28]. This discrepancy could be due to the inappropriate use of antibiotics on the farms represents a selective pressure for resistant bacteria which can develop cross-resistance between several classes of antibiotics [49].

The prevalence of MDR *K. pneumoniae*, *P. mirabilis*, and *E. cloacae* in this study was 81.8%, 77.8%, and 75%, respectively. This is in line with the study conducted in Bangladesh *P. mirabilis* 83% [41]. However, these findings are higher than a study conducted in Jimma, Ethiopia, where 57.1% of *K. pneumoniae* and 50.0% of *P. mirabilis* reported as MDR [28], and 53.57% of *K. pneumoniae* was also reported as MDR by a study from Indonesia [42]. These bacteria may develop ABR via acquired mechanisms. The acquired resistance occurs through horizontal gene transfer such as conjugation, transduction, and transformation from other resistant bacteria. Additionally, mutations in the gene could also cause this MDR when the bacteria are constantly under pressure after being exposed to antibiotics [50].

The overall prevalence of MDR Enterobacteriaceae was 116 (81.1%; 95% CI: 74.7–87.5). This is higher than a report from Jimma, Ethiopia where the MDR prevalence was 52.5% [28]. This difference may be due to the types of commercial chicken, and the number of farms included in the study [51]. For instance, the present study includes multiple poultry farms and different types of commercial chickens such as layer, broiler, and day-old chickens.

In the present study, the ESBL-producing Enterobacteriaceae from chicken droppings was 8.4%, and the prevalence of ESBL-producing *E. coli* and *K. pneumoniae* was 12.6% and 9.1%, respectively. This finding is in line with the studies conducted in Uganda *E. coli* 17.5% [52], Egypt *E. coli* 12.5% [31], Tanzania *E. coli* 10.29% [26], India *K. pneumoniae* 5% [53], and Indonesia *E. coli* 7.03% [27], and it is lower than studies in Zambia *E. coli* 20.1% [40], Nigeria *E. coli* 37.8% [54], Ghana *E. coli* 29% [44]. In contrast, our result is higher than the study conducted in Tanzania *E. coli* 4.7% [29], India *E. coli* 5.3% [53], and Indonesia *E. coli* 3.3% [55]. This variation in prevalence rates could be the difference in ESBL screening methods used and it might be due to poor animal management practices and hygienic conditions; chicken dropping has contact with chicken feedings that enhance the spread of MDR bacteria in the flock [35].

Extended-spectrum beta-lactamase-producing *E. coli* and *K. pneumoniae* have been frequently reported in poultry and therefore poultry production might serve as a reservoir for ESBL-producing strains [56]. Different ESBL genes might exist and spread on various mobile genetic elements like plasmids that can transfer horizontally between bacterial species [57]. There is a significant association between the prevalence of ESBL and flock size (p = 0.014). The occurrence of ESBL increased with high flock size than the lower number of chickens on the farm. This finding is in agreement with the results of a study conducted in Uganda [52]. High flock size may increase stocking density which led to increased levels of airborne and respiratory disease transmission, thus increasing the risk of their environmental contamination with different bacterial strains. This is probably the cause of the reduced immune responses observed at high stocking densities, as high stocking density causes reduced feed consumption and lower growth rates. This leads to more susceptibility to ESBL-producing bacterial infection [58].

In this study, chickens aged less than two months were high risk to ESBL-producing Enterobacteriaceae carriage (p = 0.037). Because the gut normal flora of these birds is still maturing, making it easy for colonization by various pathogenic bacteria if they are exposed to the poultry environment. Additionally, due to their lower immunity, survival and multiplication of ingested ESBL-producing Enterobacteriaceae via the gastrointestinal tract is increased [59].

The occurrence of ESBL was significantly associated with the use of ciprofloxacin (p = 0.001) and trimethoprim-sulphadiazine (p = 0.018). This may be because the inappropriate use of these antibiotics in the farms represents a selective pressure for resistant bacteria which can develop cross-resistance between several classes of antibiotics like beta-lactam antibiotics [49]. In addition, these antibiotics were used for the treatment and prevent diseases in commercial farms in mass with crowded poultry flocks, these practices lead to a massive accumulation of antibiotics in the farm environment and facilitate the acquisition of resistance genes in bacteria coming in contact with them [60]. These bacteria are capable of being transmitted to humans through direct contact with infected birds and the consumption of contaminated food chains [15].

## Limitations of the study

Isolation was performed on MacConkey and XLD agar which limits the isolation of fastidious Enterobacteriaceae and molecular characterization of the isolates wasn't conducted.

## Conclusions and recommendations

A high prevalence of clinically important bacterial pathogens with a high prevalence of MDR and ESBL-producing *E. coli* and K. *pneumoniae* were recovered in the present study. Poultry farms may be one potential reservoir for Enterobacteriaceae that shed into the environment through faecal matter contamination which might be a potential public health concern. Therefore, close supervision of poultry farms handling large flocks and day-old chickens should not be underestimated. The prudent use of antibiotics in poultry farms is better to be strictly supervised.

## Supporting information

**S1 File. Questionnaire English version.**
(DOCX)

## Acknowledgments

The authors would like to thank the Department of Medical Microbiology, School of Biomedical and Laboratory Sciences, College of Medicine and Health Sciences, the University of Gondar. We also acknowledge Gondar city's rural and urban agriculture centre and the poultry farm owners.

## Author Contributions

**Conceptualization:** Mitkie Tigabie, Feleke Moges.

**Data curation:** Mitkie Tigabie, Azanaw Amare.

**Formal analysis:** Mitkie Tigabie, Azanaw Amare, Feleke Moges.

**Investigation:** Mitkie Tigabie.

**Methodology:** Mitkie Tigabie, Feleke Moges.

**Project administration:** Mitkie Tigabie.

**Resources:** Mitkie Tigabie.

**Software:** Mitkie Tigabie, Feleke Moges.

**Supervision:** Sirak Biset, Teshome Belachew, Azanaw Amare, Feleke Moges.

**Validation:** Sirak Biset, Teshome Belachew, Azanaw Amare, Feleke Moges.

**Visualization:** Sirak Biset, Teshome Belachew, Azanaw Amare, Feleke Moges.

**Writing – original draft:** Mitkie Tigabie, Sirak Biset, Teshome Belachew, Azanaw Amare, Feleke Moges.

**Writing – review & editing:** Mitkie Tigabie, Sirak Biset, Teshome Belachew, Azanaw Amare, Feleke Moges.

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
