## [Decision Letter · Decision Letter 0]

13 Apr 2023

PONE-D-23-06358Multidrug-resistant and extended-spectrum beta-lactamase-producing Enterobacteriaceae isolated from chicken droppings in poultry farms at Gondar city, Northwest EthiopiaPLOS ONE

Dear Dr. Tigabie,

Thank you for submitting your manuscript to PLOS ONE. After careful consideration, we feel that it has merit but does not fully meet PLOS ONE’s publication criteria as it currently stands. Therefore, we invite you to submit a revised version of the manuscript that addresses the points raised during the review process.

We look forward to receiving your revised manuscript.

Kind regards,

Md. Tanvir Rahman, DVM, MSc, PhD

Academic Editor

PLOS ONE

Journal Requirements:

Additional Editor Comments (if provided):

Dear Authors,

Please see the comments of the reviewers and take the necessary actions.

Best wishes,

Tanvir

==

Prof. Md. Tanvir Rahman

DVM, MSc (Canada), Ph.D. (UK), Postdoc (Germany)

Department of Microbiology and Hygiene,

Faculty of Veterinary Science,

Bangladesh Agricultural University,

Mymensingh-2202, Bangladesh.

Phone. + 88-01913323307; Fax + 88-09161510

E.mail: tanvirahman@bau.edu.bd

http://vmh.bau.edu.bd/profile/VMH1005

https://orcid.org/0000-0001-5432-480X

https://sites.google.com/site/tanvirahman/Home

https://www.researchgate.net/profile/DrMdTanvir_Rahman/research

Reviewers' comments:

Reviewer's Responses to Questions

**Comments to the Author**

1. Is the manuscript technically sound, and do the data support the conclusions?

Reviewer #1: Yes

Reviewer #2: Partly

2. Has the statistical analysis been performed appropriately and rigorously? 

Reviewer #1: Yes

Reviewer #2: Yes

3. Have the authors made all data underlying the findings in their manuscript fully available?

Reviewer #1: Yes

Reviewer #2: Yes

4. Is the manuscript presented in an intelligible fashion and written in standard English?

Reviewer #1: Yes

Reviewer #2: No

5. Review Comments to the Author

Reviewer #1: 1. There are many places in the manuscript written wrongly used words, phrases and sentences. The English correction is needed for sentence to sentence.

2. The theme of the abstract is not highlighted/reflected to emphasize the title. Please modify the abstract with noting a brief primary objective.

3. The Introduction section is very weak and uninformative. Please strengthen the introduction with relevant citations. There is so much languages problem. Please modify it by Native English Speaker. What is the significance of the study? How is the study different from others?…….Please note it down and rectify accordingly?

4. Results and discussion portion is haphazardly written and difficult to correlate with each other. Please modify the discussion in systemic manner with relevant references. Please modify the discussion by strengthening the results. There is also lacuna of the representation of the discussion in the present endeavor. Please modify the discussion with proper justification and analysis by strengthening the novelty of the work.

Reviewer #2: Abstract

While the author aimed to Enterobacteriaceae, they have mentioned only enrichment and isolation of Salmonella.

They said isolation of E. coli, Salmonella and P. mirabilis. But, just next to that they mentioned 9.1% of K. pneumonia as ESBL!! Strange, why didn’t they mention K. pneumonia earlier?

Introduction

The importance is nicely described with necessary references; however, this could be further improved with information on ESBL.

Few typo mistakes.

Materials and methods

Although they targeted Enterobacteriaceae, isolation was performed on MacConkey and XLD agar which limits the isolation of other enterobacteriae and increasing the chance of missing bacteria that does not grow on those media properly. So, my suggestion is, instead of saying Enterobacteriaceae, they could mention specific bacteria like E. coli and Salmonella…

Isolation techniques are not properly described.

The authors didn’t mention, how they calculated sample size, i.e. no sample size calculation was performed!!

Line 158: 5%???

Results

Nicely presented with easy going Tables.

In Prevalence, they didn’t mention about K. pneumonia, although it’s prevalence was not that low. I think it worth to mention in the prevalence section. In addition, they also isolated E. cloacae, but no mention in the prevalence or isolation section (Table 4)

But, in Antibiotic resistance K. pneumonia suddenly came (Table 3)!!

Discussion

The authors have discussed the findings nicely with evidences from different part of the world. However, the authors have identified several general characteristics significantly associated with the occurrence of ESBL, but they didn’t discussion anything about them.

The discussion feels very monotonous. It could be improved with suggestion from a person with good English writing skill.

Conclusion and recommendations

Waste disposal was not significantly associated with occurrence of ESBL according to their findings (Table 1). Rather, some critical factors were identified, which could be highlighted in the conclusion section.

6. PLOS authors have the option to publish the peer review history of their article (what does this mean?). If published, this will include your full peer review and any attached files.

Reviewer #1: No

Reviewer #2: **Yes: **Jayedul Hassan

---

## [Author Response · Author response to Decision Letter 0]

17 May 2023

PONE-D-23-06358

Multidrug-resistant and extended-spectrum beta-lactamase-producing Enterobacteriaceae isolated from chicken droppings in poultry farms at Gondar city, Northwest Ethiopia

PLOS ONE

Dear Dr. Tigabie,

Thank you for submitting your manuscript to PLOS ONE. After careful consideration, we feel that it has merit but does not fully meet PLOS ONE’s publication criteria as it currently stands. Therefore, we invite you to submit a revised version of the manuscript that addresses the points raised during the review process.

We look forward to receiving your revised manuscript.

Kind regards,

Md. Tanvir Rahman, DVM, MSc, PhD

Academic Editor

PLOS ONE

Journal Requirements:

Additional Editor Comments (if provided):

Dear Authors,

Please see the comments of the reviewers and take the necessary actions.

Best wishes,

Tanvir

==

Prof. Md. Tanvir Rahman

DVM, MSc (Canada), Ph.D. (UK), Postdoc (Germany)

Department of Microbiology and Hygiene,

Faculty of Veterinary Science,

Bangladesh Agricultural University,

Mymensingh-2202, Bangladesh.

Phone. + 88-01913323307; Fax + 88-09161510

E.mail: tanvirahman@bau.edu.bd

http://vmh.bau.edu.bd/profile/VMH1005

https://orcid.org/0000-0001-5432-480X

https://sites.google.com/site/tanvirahman/Home

https://www.researchgate.net/profile/DrMdTanvir_Rahman/research

Reviewers' comments:

Reviewer's Responses to Questions

Comments to the Author

1. Is the manuscript technically sound, and do the data support the conclusions?

Reviewer #1: Yes

Reviewer #2: Partly

2. Has the statistical analysis been performed appropriately and rigorously?

Reviewer #1: Yes

Reviewer #2: Yes

3. Have the authors made all data underlying the findings in their manuscript fully available?

Reviewer #1: Yes

Reviewer #2: Yes

4. Is the manuscript presented in an intelligible fashion and written in standard English?

Reviewer #1: Yes

Reviewer #2: No

5. Review Comments to the Author

Reviewer #1: 1. There are many places in the manuscript written wrongly used words, phrases and sentences. The English correction is needed for sentence to sentence.

2. The theme of the abstract is not highlighted/reflected to emphasize the title. Please modify the abstract with noting a brief primary objective.

3. The Introduction section is very weak and uninformative. Please strengthen the introduction with relevant citations. There is so much languages problem. Please modify it by Native English Speaker. What is the significance of the study? How is the study different from others?…….Please note it down and rectify accordingly?

4. Results and discussion portion is haphazardly written and difficult to correlate with each other. Please modify the discussion in systemic manner with relevant references. Please modify the discussion by strengthening the results. There is also lacuna of the representation of the discussion in the present endeavor. Please modify the discussion with proper justification and analysis by strengthening the novelty of the work.

Reviewer #2: Abstract

While the author aimed to Enterobacteriaceae, they have mentioned only enrichment and isolation of Salmonella.

They said isolation of E. coli, Salmonella and P. mirabilis. But, just next to that they mentioned 9.1% of K. pneumonia as ESBL!! Strange, why didn’t they mention K. pneumonia earlier?

Introduction

The importance is nicely described with necessary references; however, this could be further improved with information on ESBL.

Few typo mistakes.

Materials and methods

Although they targeted Enterobacteriaceae, isolation was performed on MacConkey and XLD agar which limits the isolation of other enterobacteriae and increasing the chance of missing bacteria that does not grow on those media properly. So, my suggestion is, instead of saying Enterobacteriaceae, they could mention specific bacteria like E. coli and Salmonella…

Isolation techniques are not properly described.

The authors didn’t mention, how they calculated sample size, i.e. no sample size calculation was performed!!

Line 158: 5%???

Results

Nicely presented with easy going Tables.

In Prevalence, they didn’t mention about K. pneumonia, although it’s prevalence was not that low. I think it worth to mention in the prevalence section. In addition, they also isolated E. cloacae, but no mention in the prevalence or isolation section (Table 4)

But, in Antibiotic resistance K. pneumonia suddenly came (Table 3)!!

Discussion

The authors have discussed the findings nicely with evidences from different part of the world. However, the authors have identified several general characteristics significantly associated with the occurrence of ESBL, but they didn’t discussion anything about them.

The discussion feels very monotonous. It could be improved with suggestion from a person with good English writing skill.

Conclusion and recommendations

Waste disposal was not significantly associated with occurrence of ESBL according to their findings (Table 1). Rather, some critical factors were identified, which could be highlighted in the conclusion section.

6. PLOS authors have the option to publish the peer review history of their article (what does this mean?). If published, this will include your full peer review and any attached files.

Do you want your identity to be public for this peer review? For information about this choice, including consent withdrawal, please see our Privacy Policy.

Reviewer #1: No

Reviewer #2: Yes: Jayedul Hassan

---

## [Editor Report · Decision Letter 1]

30 May 2023

Multidrug-resistant and extended-spectrum beta-lactamase-producing Enterobacteriaceae isolated from chicken droppings in poultry farms at Gondar city, Northwest Ethiopia

PONE-D-23-06358R1

Dear Dr. Tigabie,

We’re pleased to inform you that your manuscript has been judged scientifically suitable for publication and will be formally accepted for publication once it meets all outstanding technical requirements.

Kind regards,

Md. Tanvir Rahman, DVM, MSc, PhD

Academic Editor

PLOS ONE

Additional Editor Comments (optional):

Thanks for addressing the comments of the reviewers and revising the manuscrot.
---

## [Editor Report · Acceptance letter]

2 Jun 2023

PONE-D-23-06358R1 

Multidrug-resistant and extended-spectrum beta-lactamase-producing Enterobacteriaceae isolated from chicken droppings in poultry farms at Gondar City, Northwest Ethiopia 

Dear Dr. Tigabie:

I'm pleased to inform you that your manuscript has been deemed suitable for publication in PLOS ONE. Congratulations! Your manuscript is now with our production department. 

Kind regards, 

on behalf of

Professor Md. Tanvir Rahman 

Academic Editor

PLOS ONE